# Technological Tools for the Conservation and Dissemination of Valencian Design Archives

**Ester Alba** [1] , **Mar Gaitán** [1,*] , **Arabella León** [1] , **Javier Sevilla** [2] , **Álvaro Solbes** [1] and **Vicente Pla** [1]

1   Department of Art History, Universitat de València, 46010 Valencia, Spain; esther.alba@uv.es (E.A.); arabella.leon@uv.es (A.L.); alvaro.solbes@uv.es (Á.S.); vicente.pla@uv.es (V.P.)

2   Institute of Robotics and Information and Communication Technologies (IRTIC), Universitat de València, 46980 Valencia, Spain; javier.sevilla@uv.es

*   Correspondence: m.gaisal@uv.es

**Abstract:** Design has shaped the world in which we live; it has improved our lives in ways we cannot even begin to imagine. Even if design is everywhere and is the memory of the world, collections associated with it are not usually well-known as design archives which capture spatial and temporal narratives. Saving these types of archives and making them accessible to the public, using them as cultural hubs, might improve our lives thanks to visual literacy, creativity, and innovation. The Arxiu Valencià del Disseny aims to recover, preserve, protect, and disseminate the archival cultural heritage of the Valencian designers. With a collection of more than 150,000 objects, it recovers history and evolution from the applied arts and product design, but with a markedly future-oriented character through the enhancement of the collections and their dissemination thanks to an intelligent computational system featuring cutting-edge technologies in order to prove our understanding of Valencian and European design. The aim of this paper is twofold: first, we introduce the AVD project, an interdisciplinary project that has been recently funded by the Valencian Government in order to preserve and promote the heritage of design archives; second, we introduce a set of interactive tools related to the project, such as the digitisation and cataloguing of the collection, the generation of a Design Memory Archive, advanced searching, and semantically relating the digitised collection of the Arxiu Valencià del Disseny based on data interoperability across its collections and a benchmarking tool for designers.

**Keywords:** design; archives; digital technologies



## 1. Introduction

The Arxiu Valencià del Disseny (Valencian Archive of Design, hereafter AVD) [1] of the University of Valencia and Escola d'Art i Superior de Disseny (EASD) aims to recover, preserve, protect and disseminate the archival cultural heritage of Valencian designers. With more than 150,000 documents in its collection, it retrieves the history and evolution of the applied arts and product design, but with a markedly future-oriented character through the enhancement of the collections and their dissemination thanks to an intel-ligent computational system featuring cutting-edge technologies to improve our understanding of Valencian and European design. The archive consists of the legacy of 24 main collections such as Lola Castelló and Vicent Martínez; Punt Mobles; Eduardo Albors; Paco Bascuñán; José Juan Belda; Pepe Benlliure; Xavier Bordils; Col·legi de Dissenyadors d'Interior de la Comunitat Valenciana—CDICV—with 15 personal archives of Valencian interior designers, considered design pioneers since 1920; Universitat Jaume I de Castelló—Dissenyadors valencians; Espai Corbat—local designs of modernist curved furniture of Viennese inspiration-; Manuel Lecuona—which provided more than 20 files from small local industries associated with product design (furniture, toys, interior decoration, glassware, etc.)-; Familia Martínez-Medina; Fernando Mateu de Ros—whose legacy

is linked to the Feria de Valencia-; Ramón Ricart Gil; Mariana Salgado; Bernardo Tejeda; Andrés Gil Cañizares; Juan Nava; Bañó & Lax Asociados, and Pedro Miralles. These legacies have been donated for public custody, preservation, and the generation of value through digital tools that can be socially useful, especially for young designers, and for the promotion of innovation in the creative industries associated with Valencian, Spanish, and international design and production ecosystems.

The legacies of Martínez Peris, Curvadora Valenciana, Gasisa and La Mediterránea, through the mediation of Manuel Lecuona, constituted the other cornerstone for the creation of the AVD, with the donation of the professional legacy of the designer Vicent Martínez in the company Punt Mobles from its foundation until 2012. In a certain sense, the history of Punt Mobles is the history of Valencian design and in its memory are present the different generations of Valencian designers and a good representation of the Eu-ropean design of the last thirty years: from the youngest to the most established such as Cul de Sac, Vicente Blasco, Manolo Bañó, Pedro Miralles, Manuel Lecuona, José Juan Belda, Isabel Martínez, El Último Grito, Jorge Pensi, Carme Pinós, Pepe Cortés, Marcelo Alegre, Alejandro Miñana, Ana Mir, Emili Pedrós, Juan Manuel Ferrero, Borja Garcia. Jorge Cortés, Rene Wansdronk, Dylon, Wheeler and Van der Broeke, and all the ex-tensive work of Terence Woodgate, Lola Castelló and Vicent Martínez. Also present are the realization of projects for public facilities such as the Planta Noble de Les Corts Va-lencianes, the Biblioteca Nacional Valenciana de Sant Miquel dels Reis, as well as projects carried out in collaboration with architects such as Rafael Moneo, Giorgio Grassi, Pedro Feduchi, Julian Esteban Chapafria, Manuel Portaceli, Carles Salvadores, Juan Añon, Gema Martí and others, for various library facilities, museum areas, and cultural centres. Also present are the realization of public facilities projects such as the Planta Noble de Les Corts Valencianes, the Biblioteca Nacional Valenciana de Sant Miquel dels Reis, as well as projects carried out in collaboration with architects such as Rafael Moneo, Giorgio Grassi, Pedro Feduchi, Julian Esteban Chapafria, Manuel Portaceli, Carles Salvadores, Juan Añon, Gema Martí and others, for various library facilities, museums, cultural, institu-tional, industrial and commercial areas. Legacies contain publications on the evolution of design in Spain during the 1980s, 1990s, and the 2000s [2,3]. Records of press releases and articles published on the designs produced, conferences held, correspondence with entities and designers, and projects for presentations at international fairs. Dossiers with technical information and planimetry design. Folders with drawings of designs. Slides and CDs with product images. All graphic editions of catalogues and communications from the background of Pam i Mig to the founding and history of Punt Mobles. The archive contains publications of all the graphic work done by graphic designers Nacho Lavernia, Pepe Gimeno, and Isabel Martínez for Punt Mobles. Along with the legacy of Punt Mobles, the Arxiu Valencià del Disseny also has, through Manuel Lecuona, the following collections: the heritage of Martínez Peris, the legacy of Curvadora valenciana, the gift of Gasisa, and the legacy of La Mediterránea. Examples are shown in Figures 1–6.

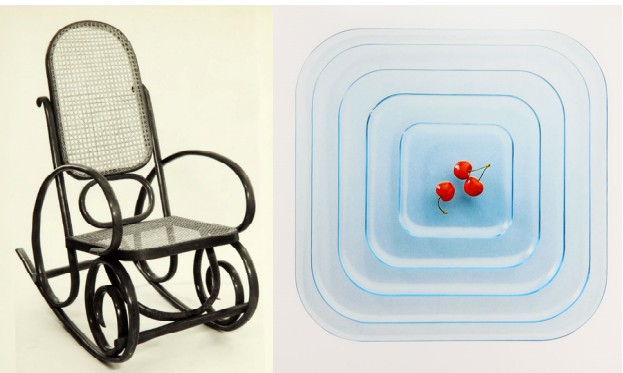

**Figure 1.** Several examples of the variety of documentation in AVD.

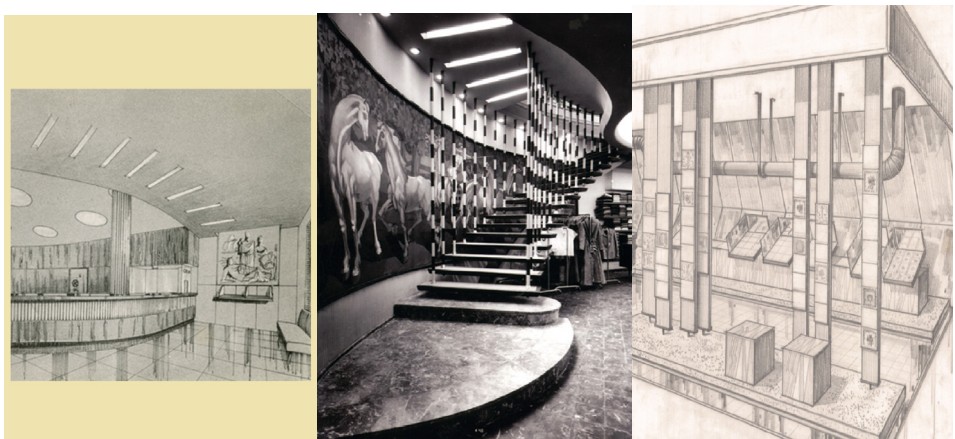

**Figure 2.** Collection College of Interior Designers. Designer Casto Buigues (1955–1971): Project for a Bank Office. Project for the warehouses of "Lanas Aragón" (Valencia) (1968). Designer José Mar-tínez Peris: Stand Project for Ceramic Company S.S.A.V. S.A. AVD.

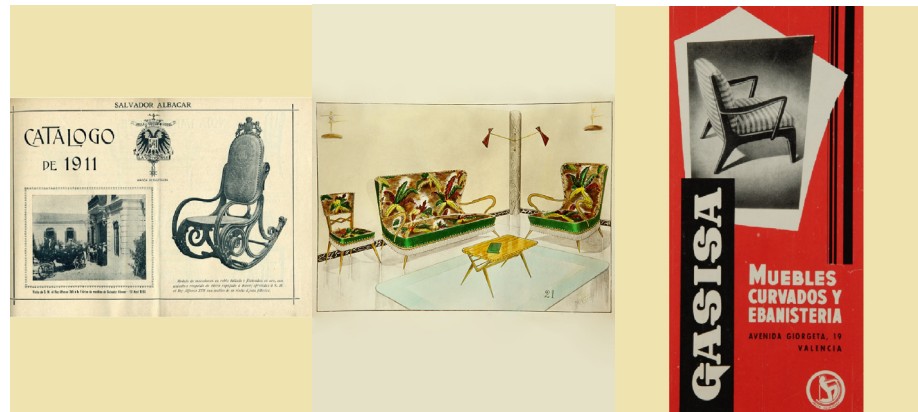

**Figure 3.** Espai Corbat Collection: Salvador Albacar Catalog (1911). Valencian Curvadora Col-lection. Gasisa Collection: Drop-down catalog of García Simón, S. A. (1962). AVD.

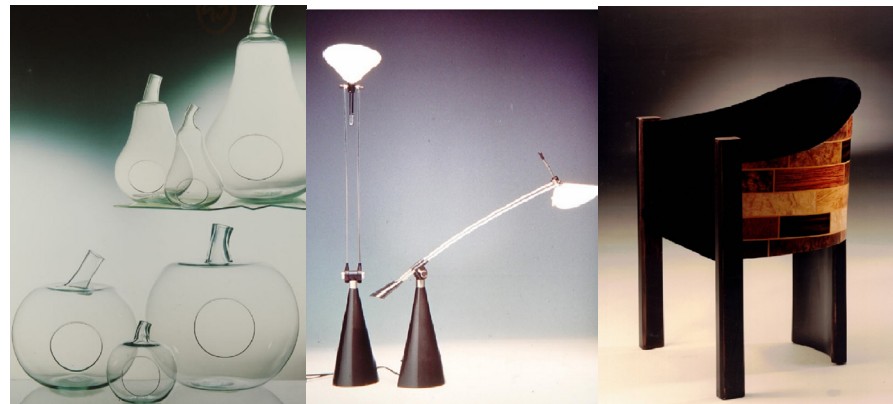

**Figure 4.** Collection Hijos de Mariano Garcia. Interior design projects (since 1920). AVD.

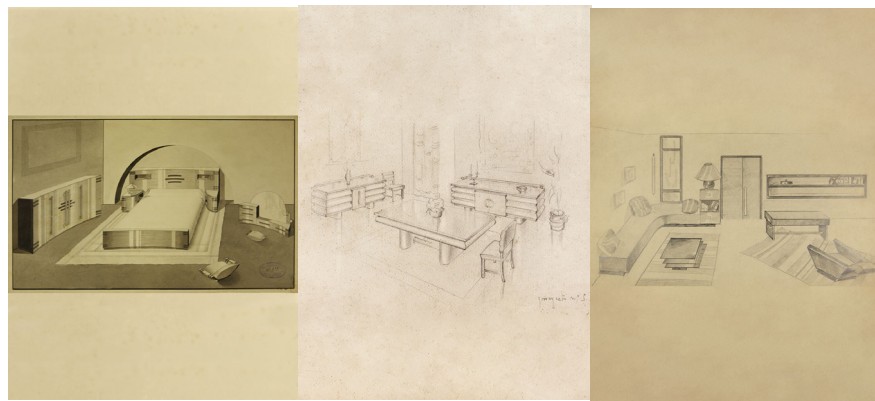

**Figure 5.** Collection Hijos de Mariano Garcia. Interior design projects (since 1920). AVD.

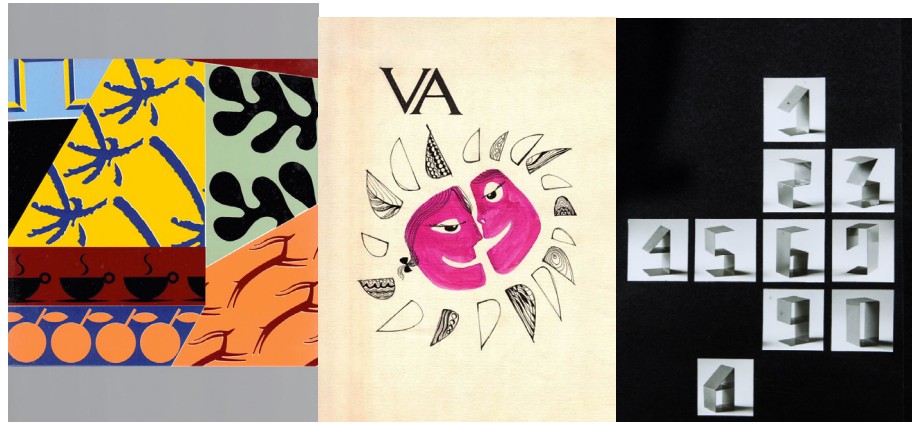

**Figure 6.** Collection of designer Paco Bascuñan (graphic design, brand, typography, and edito-rial design). Xavier Bordils Collection (industrial design and global design). Designer Vicent Mar-tínez (graphic design, furniture, and product design). AVD.

The constitution of the archive is justified by the need to create a space for the artistic legacy of the first generation of Valencian designers, which has been a historic generation for Valencian and Spanish design, fundamental to the economic modernisation of this country, committed, and generous. In this sense, the creation of an AVD responds to the cessation or donation of professional archives in a disinterested manner. In this sense, by analysing the different legal options for the contribution of works, the AVD has planned to increase its collection through two legal forms: (i) donation or (ii) gratuitous bailment.

Thus, the AVD is intended to contribute to the local design system and generate a space where all the information related to the design is in its different areas (product, industrial, graphic, communication, interior design, etc.), formats, and levels of rele-vance. The main function of the archive is to preserve the memory of the design culture in the Valencian Community (Spain) through the collection, conservation, and cataloguing of the documentation generated by the main actors of a design system (designers, design service companies, design departments/areas of the main Valencian companies, institutions promoting design, cultural activities, and actions related to design developed by different social agents to preserve this legacy). This legacy can materialise historical memory, studied, and made available to researchers on design and modern culture from original documentation on the processes of creation, production, dissemination, sale and use of objects, communications, and design services. To this end, AVD consists of a physical documentary collection (storage and maintenance of original documents) and a repository (storage, organisation, maintenance, and dissemination of documentation in digital format) to make all its contents available to the scientific community, designers, and companies, as well as to share research and collections with other similar institutions at national and

international levels. This structure aims to dynamise innovative research and creative activities in the field of design.

This will be achieved thanks to the digitisation and cataloguing of the collection (Figure 7). We will develop an exploratory web search engine capable of semantically linking digitised files and allowing their visualisation through highly interactive spatiotemporal maps and generating Product Maps to analyse the features of the products [4]. We will also apply immersive digital experiences so that users will be able to access design objects, companies' documentation, and catalogues, from which users will be able to find significant relationships between them, transforming these collections into a substantial source for the promotion and development of innovative culture [5].

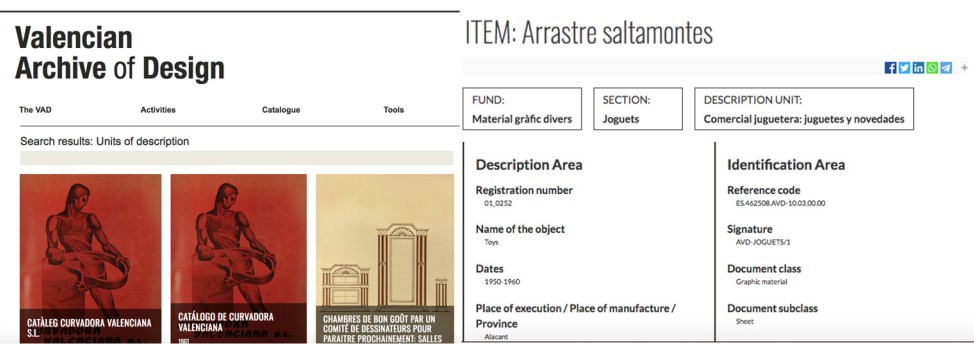

**Figure 7.** Web with AVD's traditional cataloguing system.

Hence, we will apply the full potential of design to identify the culture around these stories and strategically use those that can be part of a greener, equal, and sustainable Europe. We will connect Cultural Heritage (CH) with CCIs to offer new models of co-creation, business, and sustainability [6] by developing cultural hubs via co-creation and living heritage frameworks.

The union of CH and CCI will also be tackled by involving design students [7] in the use and reuse of tools developed for AVD, especially those related to CH. Moreover, we ensure the transfer of knowledge through alliances with other stakeholders such as SMEs and NGOs [8,9]. The results will help articulate links between companies and the public good to collaborate with the university sphere in research and analysis, and in the resolution and production of some of their needs, difficulties, or proposals in the field of design [10].

However, this design is expected to continue in the future. We will apply oriented research to create innovative processes and products with the aim of optimising resource consumption, extending the life of cycle products, and tackling problems from a design perspective [11]. These solutions include SMes, NGOs, academia (universities and design schools, especially but not limited to), the government, and other stakeholders. As such, we will address the entire project from a quintuple innovation helix framework [12,13], where stakeholder analysis and qualitative interviews will be used.

Because CCIs have improved the use of technologies such as AR/VR, AI, or blockchain [14], we take it into consideration to (1) create a Comparative Evaluation Tool through the social participation of the stakeholders involved (professionals from the sector, de-signers, industry, commerce, advertising, marketing, and the training sector); (2) develop an exploratory web search engine capable of semantically linking the digitised files and allowing their visualisation through highly interactive spatio-temporal maps and generating Product Maps to analyse the product features; (3) apply immersive digital experiences such as AR and blockchain technologies, so CCIs can offer their clients more attractive creations at the time they can fight pi-racy; and (4) implement innovative methodologies to conserve these types of objects, including material characterisation according to the latest material science that will preserve design heritage.

Finally, cultural hubs [15] will be created and will act as delocalised headquarters, following the spirit of the Bauhaus, where art, architecture, and design are present and connected with other sectors such as technology and sustainable development to be spaces of creation [16–19], without forgetting to approach gender issues and democratic values [20]. These headquarters address different societal challenges (SDGs) and design concepts, from history to technology to sustainability. In addition, it will consider a virtual sphere with physical spaces as a hybrid environment where artists, artisans, designers, creatives, and other stakeholders will interact.

As will be shown in the following sections, this archive is not a mere repository of information, but becomes a powerful source of information not only for academic researchers but also for designers. In fact, to the best of our knowledge, there is still no archive that not only reviews the documents but also analyzes the products described in them. This in itself is an innovation, since traditional archives are limited to storing documents without analysing the content, let alone putting them in context and relating them to each other. However, we wondered how to meet the demands of various sectors, for numerous tools are being developed that take into account each of the different audiences, not only that, but they are being developed and tested with them.

This study presents the rationale for this project. Section 2 explains the current state of the art, whereas Section 3 specifies the importance of safeguarding the Valencian Design and the main objectives of the AVD. Section 4 explains the current development of AVD tools. Finally, we present the conclusions regarding the project's expected outcomes.

## 2. State of the Art

### 2.1. Archives and Digital Technologies

Archives are more than a compendium. They are the product of a policy around discipline, and are themselves possessors and producers of a discourse that perpetuates and supports the discipline that gave rise to them.

The content of archives can vary greatly, containing tangible and intangible material containing part of history, heritage, or simply preserved valuable information, and, in some cases, the records can be considered "history itself".

Technologies are present in all areas of cultural heritage, including archives, and they can render archives anachronistic or obsolete. The main problem with the advent of archive technology is the time and effort required to digitise documents, which is why many archives are inaccessible online. On the other hand, this effort is worthwhile, the ease with which they can publish their own materials online, and the volume of data that can be published brings this heritage closer to the specialist and general public and facilitates the dissemination and knowledge of this heritage.

Although paper records are the most common and understood, the number of digital archives is increasing. Nonetheless, as [21] states, a digital twin is a record that comprises granular objects that are scattered yet linked across networks. Archival digitisation over the past few decades has resulted in an increase in digitally available archival data. However, some digitisation approaches fall short of the current requirements for structured, integated, interoperable, and interrogable data.

To properly address these issues, it is necessary to require specific skills not only to understand those records but also to preserve and disseminate them, especially when it comes to archives such as the AVD, which, when properly used, can act as cultural hubs and might improve our lives thanks to visual literacy, creativity, and innovation. The use of digital humanities might be a powerful way to increase the understanding, conservation, and preservation of design heritage. Digital humanities allows not only the creation of relevant databases but also the development of new methodologies that through the data obtained permit further research and new knowledge in the field of humanities. Paul Jaskot [22] distinguishes four approaches: digital storytelling including new forms of research using digital historical sources; text-based methodologies such as textual semantic analysis; network analysis connecting databases such as knowledge graphs; and spatial

analysis through digital map visualization and 3D modelling. All of these are applied to AVD.

As some literature demonstrates [23,24] digitalization enables researchers to pay closer attention to detail while still being able to abstract and generalise, contributing to a historical turn in management, making practice-oriented research easier and of improved quality, and facilitating the adoption of critical perspectives through micro-level data in the form of digital archives and online content. However, at the same time, these types of archives present many challenges in terms of data collection, such as consent, power, inclusivity, transparency, ethics, and privacy [25], especially when dealing with the work of living designers.

Similarly, the literature shows few examples of the relationship between archival description and descriptive metadata of digital objects, that is, how archivists connect archival description with digital items in the digital realm [26]. Some examples can be found in the literature [27–30]. Nonetheless, there are few examples of archives that not only describe the archival description with what is depicted in the document (referring document to any type of paper, magazine, journal, video, etc.), just the Peoples Graphic Design Archive and the AIGA Design Archive [31–33], but both fail to connect objects, authors, and companies. The AVD goes further in the archival description by cataloguing not only the document itself but also the product described on it (e.g., a chair, a toy, a lamp, etc.). This is even more challenging when proper metadata must be added to create proper relations among the objects.

In this regard, some authors [34,35] already have expressed how archives are exploring the use of the Semantic Web and its related technologies to enhance the management, accessibility, and interoperability of their collections. By performing queries across distributed data sources, enabling researchers and users to seamlessly access and combine information from multiple archives by publishing their metadata and contending as Linked Open Data, enriching metadata to provide additional context and knowledge about their resources, and improving the discoverability and understanding of their collections, the semantic web enables long-term preservation purposes. In fact, the Association of Research Libraries (ARL) recognizes the importance of using linked open data (LOD) to describe and connect resources, to mutually enrich Wikimedia and library discovery sources, and at the time they urge to establish learning communities for Wiki-medians in libraries, cultural heritage, and research institutions to use Wikimedia projects to address cultural barriers in traditional library and archival practices [36].

In this regard, one of the main changes that digital humanities have introduced is the concept of public use, which makes history more accessible by linking data into space-temporal frameworks [37]. Obviously, in every open-access project, access is fundamental, but when it comes to designing archives even more, they can act as incubators for future designers, not only to get inspired but also to understand the processes that past designers have taken to create any sort of product. In this sense, archivists are increasingly paying attention to access, cross-disciplinary collaborations are essential [38] and collaborations between humanities and computer scientists can be fruitful. AVD is not strange to this interdisciplinarity. We are a team of art historians, historians, de-signers, geographers, and computer scientists. AVD, as a design archive, aims to have a transformative impact on the way archival materials are preserved, managed, accessed, and shared, thanks to the use of digital technologies that have revolutionised the archival field, offering numerous advantages and opportunities, as explained in this paper.

### 2.2. Design Archives

Andrea Giunta [39] proposes the archive as a site of knowledge construction, but what are the places of the archive in design? Archives are the product of a series of conservation policies and the memorisation of events, but they are mainly producers of knowledge [40]. It is essential to revise the idea of archiving and to understand it as a new medium, which is

not only about researching existing documents but also about promoting and constituting new digital and accessible archives.

From archives in general, and from design in particular, it is possible to contribute to the resumption and sustenance of creative forces through research to trace other histories from the revision of the cultural past, but also from the awareness of the present. This is why the digitisation and dissemination of design archives are essential to raising awareness of a discipline that is intrinsic to civil society, which is consumer culture, and is constantly evolving.

When we consider an archive, easily archivable items or objects come into mind. The reality is that, especially in a design archive, the variety of formats and nature of the work make this task difficult. A wide variety of products and processes can be archived, ranging from the first sketches of a project to the greatest achievements in the history of design, material objects, virtual developments, two-dimensional elements, three-dimensional elements, documents, and ideas. The size of the archive can vary as well as the format; they can be photographs, digital memory, or a sketch of something ephemeral.

Design archives can be understood from a museum, artistic, design, or business point of view, which makes their conservation, especially their use and dissemination, difficult, as they can be aimed at several audiences simultaneously. The systematisation of information facilitates the conservation and dissemination of archives, as it constitutes an important tool in the recovery of memory, given its association with history.

Therefore, they are useful for designers and cultural industries and contribute to the development of collective awareness of the discipline and its achievements. They are the letters of the introduction and the primary reference material in the creative process. In terms of design education, it is easy to identify several relationships; for example, archives created in the classroom are the perfect laboratory for motivating rigor, which will later be useful to professionals.

The There are some examples of design archives with online repositories, such as archives design [40,41] which is a digital archive of graphic design-related items that are available on the Internet which is a digital archive of graphic design-related items available on The Internet Archives. Another example is The People's Graphic Design Archive, which is a crowd-sourced virtual archive of inclusive graphic design history. The Archive includes everything from finished projects to processes, photos, correspondence, oral histories, anecdotes, articles, essays, and other supporting material. The Architectural Design Archive [40] is another type of digital repository that focuses on architecture and provides a digital platform to support students throughout their degrees, after graduation, and beyond. This innovative project archives creative practices and showcases them globally, offering both local and international reach. This archive also provides a record of the best works by students of the DPA Architectural Design Department at the ETSAM School of Architecture, from the first undergraduate course to the graduate master's degree. Sol Design Archive [42] is an online archive of design with a focus on 20th-century Portuguese graphic design and illustration but a voracious appetite for the full breadth of visual expression. Design Reviewed [43] is a personal project dedicated to digitally preserving graphic design history and documenting the vast visual culture from the last century. The Bauhaus Archive Museum of Design [44] which provides a digital collection of documents, photographs, and architectural plans related to the Bauhaus movement. This online archive allows users to explore the history and influence of influential design schools. ArkDes [45], the Swedish Center for Architecture and Design, contains archives of 600 architects in Sweden. More generally, the European Art and Design Collection [46] provides access to millions of digitised cultural heritage objects, including design-related materials from museums and archives across Europe.

These examples demonstrate how design archives leverage digital technologies to make their collections more accessible, engaging, and relevant to broader audiences. By embracing digital tools, design archives can preserve their materials, inspire new creativity, and contribute to the study and appreciation of design histories. Nonetheless, there is

a lack of academic publishing on these and other examples that can encourage research, particularly when it comes to applying digital technologies to design archives.

## 3. Safeguarding Valencian Historical Design through the Development of a Design Archive

In Valencia, design history can be traced from the Escuela de Artes y Oficios de Valencia (School of Arts and Crafts) which was created in 1850 as a response to the lack of training in the artistic trade of Valencian craftsmen in the second half of the 19th century. Under the auspices of the San Carlos Academy and the School of Fine Arts, it focused on training working classes in applied arts to generate top-level professionals to supply the incipient Valencian industry. This led to the creation of an extensive artistic and cultural background that shows the aesthetic evolution of the Valencian craft world and applied arts. It also demonstrated the assimilation of new techniques and typologies taught by the School of Arts and Crafts, with examples that take us back to the Arts and Crafts movement, modernism, and the rise of the avant-garde. This school is the predecessor of the current Design School which currently trains Valencian designers.

It was until the sixties when interest in design and its association with the industry's productive modes emerged in Valencia. As in other Spanish cities (mainly Madrid and Barcelona), architects were concerned about its implementation. Thus, in 1967, the first edition of the Conversaciones sobre Diseño Industrial (Industrial Design Conversations) was held at the Valencian Architects Association with professionals such as Emilio Giménez Julián and art critics such as Tomàs Llorens Serra (director of the IVAM, Museo Nacional Centro de Arte Reina Sofía, and Museo Nacional Thyssen-Bornemisza). In these conversations important personalities participated such as André Ricard, Miguel Milá or Tomás Maldonado, president of the ICSID and former director of the prestigious Hochschule für Gestaltung in Ulm founded by Max Bill. Among the contributions to the Conversaciones, Alexandre Cirici deserves a special tribute since it presented an early but in-depth analysis of the main factors involved in designing, from the intermediary to the user, the designer, the role of public institutions, and technology. Cirici also introduced critical views on the groundbreaking implementation of computing resources in that time design.

In 1984, the key role of design was politically acknowledged by the Institute for Small and Medium-Sized Enterprises (IMPIVA). As a result of this policy of industrial support, professionals and groups of designers emerged, all of whom were awarded the National Design Prize by the Spanish Ministry of Culture, including Javier Mariscal (National Design Prize by Spain, 1999), Vicent Martínez (National Design Prize by Spain, with Punt Mobles, 1997), Lola Castelló (founders of Punt Mobles), Nacho Lavernia (National Design Prize by Spain, 2012), Pepe Gimeno (National Design Prize by Spain, 2020), and Marisa Gallén (National Design Prize by Spain, 2007). Since the mid-1980s, the work of these pioneers has been reviewed through various exhibitions, including: Diseño en España (Brussels, 1985), Diseño industrial en España (Madrid, 1998), Signos del siglo (Madrid, 2000), and the more recently, Suma y Sigue del Diseño in the Valencian region (Valencia, 2009). These exhibitions have produced significant publications that will allow researchers to deepen their understanding of Spanish design, Europe, and the rest of the world. It is precisely because of the discipline's short history that research work has yet to be carried out on, among other issues, the presence of women in design or their relationship with other contexts, such as how it was born and developed. This is a necessary task that should be undertaken as soon as possible.

In this context, the AVD arose as an initiative of the Universitat de València and the EASD València [47], thanks to the donation of the professional legacy of the designers Vicent Martínez and Lola Castelló, as well as the documentation associated with Punt Mobles, as well as their personal archives, with their design output as creators. The AVD also holds the memory of several important regional and European designers of the last 30 years, from the youngest to the most consecrated, such as Cul de Sac, Vicente Blasco, Manolo Bañó, Pedro Miralles, Manuel Lecuona, José Juan Belda, Isabel Martínez, El Úl-timo Grito,

Jorge Pensi, Carme Pinós, Pepe Cortés, Marcelo Alegre, Alejandro Miñana, Ana Mir, Emili Pedrós, Juan Manuel Ferrero, Borja García, among others. The importance and variety of collections are among its strengths. It integrates the archives of several pioneering designers in the field of Spanish design, as well as several collections from small- and medium-sized industries that turned product design and graphic design into their identity. These records have the importance of not focusing the attention of the design on its final product, but to integrate from the previous conceptual drawings in which the idea of the product or graphics is developed, the different models designed, prototypes, to the documentation that is generated later in the process of launching, marketing, and commercialisation of the products: company catalogues, materials for fairs, photos that include diverse materials (acetates, slides, commercial photographs, etc.), drawings, elevations, plans, and interior design sections, and company sample catalogues (fabrics for furniture, toys, home design, furniture), up to prototypes. It should be emphasised that this diversity and complexity of the materials to be preserved in the AVD requires an additional effort in the processes of cataloguing and digitisation of the collections, as well as in the decisions on the appropriate documentary process for their physical and digital custody.

Moreover, the archive is also custodian of public projects such as the design of the Noble Plant at Les Corts Valencianes, the Valencian National Library in Sant Miquel dels Reis, as well as projects carried out in collaboration with architects such as Rafael Moneo, Giorgio Grassi, Pedro Feduchi, Julián Esteban Chapapría, Manuel Portaceli, Carles Salvadores, Juan Añón, Gema Martí and others, for various library facilities, museum and cultural areas, institutional, industrial and commercial. It also contains publications related to the evolution of design in Spain during the 1980s, 1990s, and the 2000s, including press releases and published articles on the designs produced, conferences held and correspondence with entities and designers, as well as projects in international fairs, dossiers with technical information and planimetries of designs, folders with drawings of designs, slides, and CDs with images of the products. All graphic editions of catalogues and communications from the background of the Pam i Mig Company to the foundation and history of the Punt Mobles Company. The archive contains publications of all the graphic works done by graphic designers Nacho Lavernia, Pepe Gimeno, and Isabel Martínez for Punt Mobles. Thanks to Manuel Lecuona, the AVD is also protecting the collections of the Martínez Peris archive, La Mediterránea, La Curvadora valenciana, the archive Hijos de Mariano García (Figure 5), the Martínez Medina archive, the Mariner archive, the IMPIVA project archive and a diverse collection of diverse graphic material.

Thanks to its twenty-four legacies at the present time, some of them encompass works and records from several professional designers or different companies, and the AVD holds the best-supplied and non-related to a museum body of documentation about design in Spain. It can provide for the general public and researchers thousands of pieces of evidence from the beginning of the Valencian furniture factory system more than a century ago (for example, Ventura Feliu) and the transformations inside pottery and glass industries (for example, La Mediterranea), shoes (e.g., Panama Jack), lighting (e.g., Lamsar), toys (e.g., Feber), and even fashion (for example, Tráfico de Modas) to the latest creations belonging to various fields of design as the institutional advertising posters campaigns and the branding and packaging most original solutions. Thanks to such a wide range of documents from many sources and authors and from different historical periods, the AVD can enlighten some key modernisation processes of the Valencian economy and can model the role of design strategies in the system of innovation. In addition to accountability records, brochures, catalogues, samples, and briefing reports, it is also worth mentioning that a significant part of the documents refers to the work-in-progress phases of design: handmade drafts, sketches, and drawings prior to the definitive creations. Therefore, AVD legacies are especially valuable in documenting design thinking.

The AVD is co-headed by two representatives, one on behalf of the Universitat de València and the other of the EASD València. The AVD organisational structure is supported by a Technical Advisory Council, in which the power to approve the reception of new

donations and to set the programs of action lies, and a Monitoring Committee, comprising all the donors of the legacies to ensure that their links to the archive remain active. In the Technical Advisory Council, in addition to the two heads and two design experts, the Deputy Chancellor of Culture and Society of the Universitat the València, the Deacon of the Geography and History College, and the CEO of the Universitat de Va-lència's Fundació General also play a decisive role. The AVD is also provided with technical staff in charge of receiving, cataloguing, digitalising documentation, managing the website, and running dissemination activities such as conferences, exhi-bitions, seminars, webinars, workshops, and publications.

Beyond the internal procedures and daily routines, AVD has among its main features setting and keeping a series of external links and collaborative ways with other institutions related to design and innovation areas. The Valencian Innovation Bureau (AVI) provides a collection for AVD and surveys its activities. The Valencian Professional Designers Asso-ciation (ADCV) and Valencian Foundation of Design (FDCV) offer sustained and mutual support to AVD to share projects and establish their common goals. Other companies, cooperatives, studies or independent professionals of design, people involved in CCI, research institutes, museums, and entrepreneurs are closely related to AVD. This growing network allows AVD to increase its collection by attracting new donors. Donations for AVD are usually implemented via permanent donation contracts and seldom through tem-porary deposit contracts. In reciprocity with the agents of the productive system around design, the AVD gives them a sustained assessment on keeping, collecting, maintaining, and managing documentation related to design.

*Aims and Scope*

The AVD far from being a traditional archive, it wants to act as a laboratory of innovation and creativity [48]. A place to preserve, protect and disseminate the legacy of Valencian designers, the history and evolution from the applied arts and product design, with a markedly future-oriented character through the enhancement of the collections and their dissemination thanks to an intelligent computational system featuring cutting-edge technologies to improve our understanding of Valencian and European design. However, it also aims to become a national and international reference for future generations [49], an engine of economic and sustainable development for the Valencian industry and an agent of innovation [50]. Our aim is to become an international reference for future generations, an engine of economic and sustainable development for the European industry, and an agent of innovation [51]. Acting as a cultural hub design will become a potential resource for innovative economies [52], through sustainable development and the promotion of creative industries [53–55].

In addition, AVD follows a people-centred conservation approach [16] requires actions where community is actively involved [56]. In this sense, AVD is already tackling young designers as some of its main stakeholders [57]. Design students can use archives as inspiration for creative and innovative design solutions [58]. Cultural heritage can provide a rich source of inspiration for designers as it reflects the history, traditions, and values of a particular culture or community. By exploring cultural heritage, designers can gain insights into the unique perspectives and experiences of different groups of people and use this knowledge to create designs that are culturally sensitive and relevant. AVD contains a wealth of historical and cultural information, which can inspire and inform design ideas, from research and inspiration to exploring the original context and intent behind specific designs or even developing innovative solutions for the storage, display, and packaging of archival materials that prioritise sustainability and conservation. Similarly, the outcomes resulting from AVD will be transferable to different agents of Valencian and European designs, where training and support actions will be developed. To this end, co-creation design laboratories will be developed with young designers to show that incorporating young minds can produce tangible results, such as innovation, economic productivity, and

social cohesion. This disruptive perspective draws from the philosophy of Bauhaus as a permanent space for teaching and innovation.

It is commonly accepted that museums act as repositories for human memory in many different fields, such as cultural, historical, and social. Nevertheless, memory is selective: when records are viewed, selected, and exhibited, a large number of people are often excluded [59]. Therefore, from a committed perspective, when we talk about sustainability, and therefore about the SDGs, it is important to include the gender perspective [60]. Current heritage studies raise the need to approach study and practice from an ethically committed perspective that includes values such as participation, cooperation, dehierarchization, multiple knowledge, and the distribution of the concept of authority. Working based on the enhancement and recognition of design heritage also implies a dehierarchization of values by including design as an essential dimension of cultural contribution in artistic and creative processes. This new approach is fundamental if we consider the role of museums and archives as agents of social transformation and creators of values, in addition to their fundamental function as a place to collect, conserve, and exhibit cultural goods. New museology and critical museology, together with feminist curatorial practices, have renewed cultural practice and forced us to see the archive and museums as a space for reflection, debate, the promotion of critical thinking, and the visualisation of subjectivities or exogenous discourses [61,62]. This change involves understanding the museum as an agent of social transformation, where re-reading the collections from a gender perspective is fundamental to this notion of social responsibility. In fact, it is in the decorative arts and in design, far from the readings of great art, where we can find and rescue the number of women who played an essential role in the field of design, both as creatives and agents of innovation, or as art or industrial directors, as their fundamental role in the context of small- and medium-sized family enterprises. From this ethical point of view, AVD safeguards heritage collections of archives of creatives, designers, and companies with the will to preserve, disseminate, and value this contemporary heritage, but at the same time make it available to current creators, creative industries, and young designers. It is at this point that valuing the work of female designers is also a substantial point, generating references, but also breaking with established social and cultural canons.

On the other hand, to achieve the above, the AVD will use a set of tools based on Archive's collection. First, we aim to maximise the potential of cultural heritage design as a driver of innovation and sustainable development. This objective is achieved by (1) providing semantics to digitised heritage objects, enabling links between different collections. The computational representation of narratives will be used to model events in space and time using highly interactive and intuitive spatial and temporal maps. We will model causal dependencies in an appropriate context, such as the Industrial Revolution and the Arts and Crafts movement, the life of a person, or the history of a community. (2) Presenting a cultural context through narratives explaining its elements. We capitalise on semantic associations to fuel narrations that: (a) elaborate on history, memories, and values rendering cultural items significant for a social group, to support their preservation and presentation; (b) Illustrate how CH is re-interpreted over time; and (c) enable exploration of knowledge and heritage [63].

Second, we will improve the understanding, conservation, and preservation of Valencian design heritage. This will be achieved by: (1) keeping alive the design ar-chives through the website, by an interactive and open way through the project website which follows the WCAG guidelines [64] and the WAI-ARIA recommendations; (2) interpreting verbal and visual content (Transcription & Articulation), understanding their format and language is mandatory. Advances in OCR, speech recognition, content-based image re-trieval, document analysis, and automatic translation simplify the process; and (3) make digitised heritage objects more accessible in an innovative way. We will elaborate on the representation or symbolism of the maps of the digitised objects and the semantic relations between them. Thus, the user can analyse objects of the same type, material, authors, and so on. We will provide online visitors with a virtual representation of these archives

through a Virtual Gallery solution; and (4) create a design memory archive that rescues the social function that design has had among the years, as well as how it has helped us improve our ways of life, hence contributing to social inclusion and sustainability. We will raise awareness of the value of culture as the fourth pillar of sustainable development.

Finally, innovative tools will be created for designers. This objective can be achieved by (1) deepening the knowledge of the relationships between user needs and business strategies. We will analyse research techniques and strategies used in national and international fairs through work with panels of designers and end-users, (2) involving young designers in the project activities, and (3) bridging the gap between designers and social inclusion. We will ensure that gender, climate change, migration, and differently abled persons are considered at every stage of the project. This will be achieved through capacity building for designers, participatory design through a people-centred approach, and inclusive design. According to the Universal Value N2. of the UN: Leave no one behind (LNOB).

In short, the AVD stands as a centre for the preservation of the important legacy of professional Valencian design, but acts as a cultural hub thanks to the digitisation of their collections and the development of specific tools, making it a potential resource for innovative economies through sustainable development and the promotion of creative industries and Valencian industrial factories. This is aligned with the current thoughts of the European Union, which will act as a new cultural project blending design and sustainability.

## 4. Outcomes of the Project

The technological tools being developed under this project cover the areas explained in the following sections.

### 4.1. Research

4.1.1. Design Memory

Research on AVD has not only focused on the digitisation and cataloguing of archival documentation but also on collective and individual memory research [65]. Semi-structured interviews were conducted with Valencian designers to investigate the peculiarities of the behaviour, relational, and cultural needs of the design community. The objective is to create an archive of memory, so-called Design Memory, through a methodology that can serve as a construction of innovation by emphasising the understanding of subject/company relational structures [66].

Within the framework of qualitative methods, the interview is similar to the technique within the framework of qualitative methods, and the interview is similar to the observation technique, particularly for the participant type. Its application is oriented towards the collection of "unstructured" data, that is, data that have not been coded at the point of collection from the perspective of a closed set of analytical categories [67]. One of the most important characteristics of qualitative techniques is that they go beyond the collection of information; that is, they try to capture the meaning that people give to their actions, ideas, and views of the world around them. While this is true, Design Memory is relatively distant from this ethnographic vision of data collection, as we strive to codify part of the information to add it to the semantic web [68]. These data are connected to the ontology so that intelligent searches can be performed with satisfactory results. Moreover, the Design Memory has a section that allows users to see the relationships generated between the data extracted from the interviews and the data of the AVD ontology [69]. This means that they could relate companies to products, people with companies, and the roles of these people, even family and work relationships, to maintain a high level of data connection. The aim is for these relationships, which have been studied at different levels, to be accessed intuitively and simply.

Although the economy of culture and experience is in many respects difficult to measure or evaluate, the role of design in the preservation of identities and the dissemination of culture and knowledge is highlighted [70]. Therefore, the differentiation of products and

services in a global market will not only be based on technological innovation, but also on the cultural value of the brand, represented by its aesthetics, significance, and functions to a lesser degree. Therefore, design enables strong links and emotional experiences to be established between brands and consumers. On the other hand, it also creates a type of association that strengthens cultural identity by making end users and consumers identify products and brands with the culture of the area in which they were created. Among the objectives of most design policies [71] in developed countries is to strengthen the design research sector and exchange knowledge through the establishment of international networks. AVD will play an important role in achieving these objectives by providing an analogue and digital space for exchanging information, identifying models of good design, promoting them through exhibitions, getting to know their authors, establishing contacts with counterpart centres in other countries, and so on. To do so, we applied ethnographic analysis to detect and identify the uses, customs, and cultural manifestations of the designer community in its relationship with artefacts, users, communications, and design activities. Memory research, as a knowledge production process, is the result of research on designers as promoters of innovation. It is a tool based on interviews with reference designers, which can be used later in the professional field by humanists, sociologists, economists, researchers, designers, and other creatives, and within the academic field by students and teachers of the degree and postgraduate design courses, with the aim of observing the variables of innovation within the context of designs, projects, create patterns, and structuring the practice of product-services design.

As such, the Valencian Design Memory (Figure 8) will become a "way of seeing" and describe reality that can hardly be traced back to a procedure that can be described in stages. This allows us to develop a descriptive and holistic vision of designer activities in their contexts. To do so, the interviews identified five identical sections: (1) Training period, which corresponds to when and how designers were formed, including their references both in designers and in movements that have influenced them the most. (2) Beginning of Professional Activities In section, we intend to see how they progress from being design students to professionals, how they enter or create their first studios, and their first projects. (3) Professional maturity. This is one of the sections from which more data can be extracted automatically because we asked about the materials, techniques, and processes they have used in their professional development, which allowed us to connect these data with those catalogued in the AVD and with the thesaurus itself. (4) Design Professions. This section was the most open, as it was adapted to each interviewee. (5) Design education. In this section, we discuss when and how they started their activities as teachers (if so).

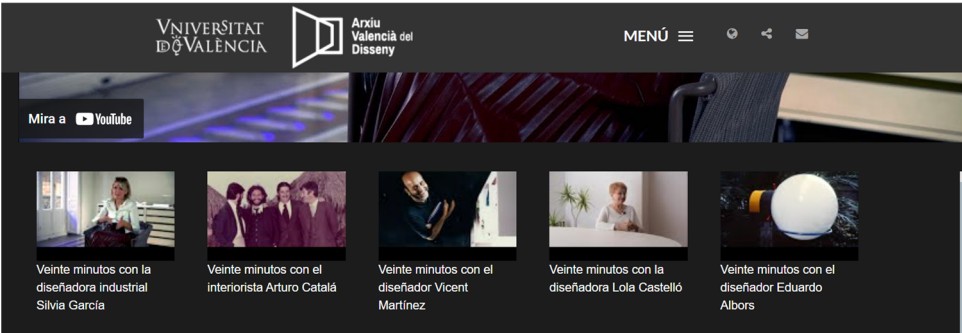

**Figure 8.** Memory Design. Recovery of the memory of women in design culture Interviews with Silvia García, businesswoman and artistic director of La Mediterranánea, and Lola Castelló, product designer and furniture designer for the company Punt Mobles.

4.1.2. Thesaurus

AVD is developing a design thesaurus that will help standardise the vocabulary used in design. A thesaurus is generally defined as a controlled vocabulary with a semantic network of unique concepts [72]. These concepts were controlled for in this study. For example,

each Preferred Term (PT) represents a single concept. TPs are ordered by hierarchies, where a Specific Term (NT) is a Generic Term (BT). TPs can also exhibit associative and equivalent relationships. The cultural heritage domain is characterised by large, rich, and heterogeneous datasets [73]. Cultural heritage institutions such as GLAMs strive to obtain controlled vocabulary based on their own collections. Some examples of cultural heritage vocabulary standardisation and its typologies can be found in [74–78]; however, to the best of our knowledge, there are no specialised designs.

To create this, we followed the ISO 25,964 international standard for the creation of thesauri and the standard established by the Getty Institute. In the case of the AVD thesaurus, the facets are those of the AAT thesaurus [79], however, in terms of concepts we are including specific ones related to design, such as product, fashion, interior and graphic design, to mention a few. This also allows cataloguers to accurately describe the product and each component, thereby facilitating the tasks of the semantic web.

Finally, as per the platform, it is being developed in VocBench, a multilingual collaborative web development platform for the management of controlled vocabularies such as ontologies (OWL), thesauri (SKOS/SKOS-XL), lexicons, and generic RDF datasets. VocBench is designed to help public administrations maintain and publish controlled vocabularies in an open and interoperable manner.

The thesaurus will be an additional tool that will be integrated into the semantic web and will serve to structure it, but it will also serve to make an exhaustive cataloguing of pieces; in other words, all inventory cards will follow the thesaurus to ensure that they are governed by the structures of the hierarchy that we have established, but also that the appropriate terminology is used. The priority is for the thesaurus to have the maximum number of terms used in the cataloguing cards; therefore, it will be work that can be expanded as the collection of the AVD expands.

*4.2. Non-Invasive Techniques for Conservation and Digital Documentation Process*

The digitisation and cataloguing of AVD collections are fundamental premises as one of the pro's identities. To achieve this, a working methodology was established in 2022, which included the necessary fields for correct cataloguing but also served to extract and include them in the knowledge graph (KG) [80]. The first step consisted of accessing the archival catalogued collections to understand the contents of the collection and to prepare an inventory card of the different products from a more patrimonial point of view and with the end user in mind.

These catalogues illustrate products of all kinds: chairs, sofas, sofas, couches, armchairs, armchairs, coat racks, and furniture. Cataloguing was carried out in a web application based on WordPress, where the files contain basic data such as registration number, secondary denomination, authorship referring to the designer or designer of the product, producer, or manufacturer of the product, as well as the dates of design, production, and exhibitions at events or fairs. One of the objectives was to try to standardise as many fields as possible using our own thesaurus of design-related terms as well as other tools, such as Geonames.

To do this, it starts with the traditional filing system, based on archival methodology, where there are only documents and their characteristics, to a model with information from other entities related to the design products that appear represented or are related to the documents. These entities are composed of a high level of products, as well as of the people and/or entities that have manufactured or designed these products. The objective is to use this model to catalogue data on documents and design products related to AVD. The model is represented by an ontology because the objectives of the project are to share the data both with people and with external computer services; thus, we work with the semantic web model and metadata mapping [81], as shown in Section 4.3.1.

However, as previously mentioned, AVD contains a wide variety of design objects. These include photographs, posters, disc covers, toys, and other artefacts that have special conservation needs beyond graphic documents. To improve the understanding of these

artefacts and to assist in their conservation, non-invasive analytical techniques are used to characterise the materials that make up them. This characterisation is essential for the proper management of preventive conservation or restorative interventions, provides data for the historical evaluation of collections, and serves as a source of inspiration for design creators.

In addition, the unique nature of some of these objects requires the use of non-destructive analysis techniques and advanced imaging technologies such as multiband imaging (MBI) and hyperspectral imaging (HSI).

Imaging techniques have undergone major developments in the 21st century [82], owing to advances in the digital age and the introduction of silicon-based sensors (CCD and CMOS). Multiband imaging allows the acquisition of several scientific photographs of cultural heritage in different ranges of the electromagnetic spectrum, mainly in the ultraviolet, visible, and near-infrared (UV-Vis-NIR) bands. The recorded images provided qualitative information about the materials and their states of conservation. In essence, images taken in the visible bands of the spectrum provide information on the surface and structural level, whereas in the non-visible bands, such as the UV, it is possible to characterise materials, especially organic compounds, based on the phenomenon of fluorescence or luminescence [83]. For example, in IR, it is possible to observe underlying layers in the case of multilayer artefacts.

The Hyperspectral Imaging has recently been used for the conservation of cultural heritage. HSI has several advantages over the traditional techniques. These include the portability of the equipment, which allows work to be carried out remotely or on-site, and its versatility in terms of the types of data acquired and processing capabilities [84]. HSI technique combines the spatial and spectral dimensions of a two-dimensional (2D) image. This depends on the resolution of the instrument. Generally, spectral information is captured in the Vis-NIR range, but equipment is also available in the UV and short-wave infrared (SWIR) ranges [85]. To do this, the system scales the amount of incident radiation or light reflected by the materials being analysed. This was translated into a reflectance curve for each captured image pixel. With data in both spatial and spectral dimensions, it is possible to apply imaging analysis methods to map regions of interest or to classify materials on a chemical basis [86].

In the case of AVD, the application of these non-invasive techniques will be focused on the one hand, on classifying and identifying pigments in polychrome artefacts of artistic value and, experimentally, on classifying deterioration of the paper support, such as foxing, microorganisms and photo-oxidation processes. Eventually, these techniques and methods will be applied to identify and preserve synthetic polymers in diverse design artefacts. Finally, in addition to contributing to the conservation of AVD, the expected results should have an impact on young design creators in the sense that material characterisation generates new motivations for product design.

*4.3. Analysis and Conceptualization: Advanced Semantic Data Visualization Tools*

In AVD, a set of semantic data visualisation tools is developed by extending the spatiotemporal visualisation ontology (STEVO) ontology [87] and its viewer. These tools link the formal definition of visualising information with the data in a knowledge graph. The viewer tool followed the STEVO content to define how to visualise the data. It also defines how the end user interacts with data. The visualisation tool of STEVO is based on the WebGan open, cross-platform, royalty-free web standard that provides an API based on OpenGL ES [88] hrough the Canvas element of HTML 5. The WebGL API is based on the components used to transform vertices or colour pixels (shaders). The constructs are semantically very similar to the OpenGL (Open Graphic Language) ES API. This API is supported by major browsers in their versions for different operating systems; Safari, Chrome, Edge, and Firefox are all members of the WebGL working group [89]. The WegGL open-access platform was compared with other similar platforms that allow graphical representation and interaction through web applications. Table 1 shows the results of this

evaluation, taking into account the possibility of displaying 3D/2D scenes, displaying maps only, open access, multiplatform implementation, adding plug-ins, and allowing 3D low-level management. Considering the table features, the best libraries are WebGL Three.JS or Unity3D, which are based on WebGL. The ThreeJS library was selected to achieve the goals of AVD projecting the Three.JS library.

**Table 1.** Features of web applications that allow graphical interaction and representation where red correspond to those features that the library has not and green the one that it has. Adapted from [90].

| Library | 3D/2D | Only Maps | Open Access | Multiplatform | Plugins | Low Level |
|---------|-------|-----------|-------------|---------------|---------|-----------|
| WebGL | ✓ | ✗ | ✓ | ✓ | ✗ | ✓ |
| D3.JS | 2D | ✗ | ✓ | ✓ | ✓ | ✗ |
| Three.JS | ✓ | ✗ | ✓ | ✓ | ✓ | ✗ |
| Unity3D | ✓ | ✗ | ✗ | ✓ | ✓ | ✗ |
| Leaflet | 2D | ✓ | ✓ | ✓ | ✓ | ✗ |
| OpenLayers | 2D | ✓ | ✓ | ✓ | ✓ | ✗ |
| Cesium.JS | 2D | ✓ | ✓ | ✓ | ✓ | ✗ |

The visualization tools are:

### 4.3.1. Product Map

A product map is a tool that helps designers represent and identify concepts derived from purchase behaviour or the use of a certain material or morphology. AVD will improve the current product maps using their collections.

This tool creates a board that visually reflects the current state of existing products in the market, either from the previous or current stages. It is a space where solutions can be contrasted and adapted to changes, and the product can be continuously improved.

Within the collections collected during the project, Product Maps constitute a search without a filter, devoid of a pre-established direction and a coded model of action. The essential idea is to identify and visualise the rhythms of innovation given that relational structures with demand can offer a range of promising areas to activate reflections in the project. Owing to these circumstances, there are no specific tools to collect information; rather, certain tools are required that are typical in the tasks of collection and classification of stimuli. There is a consolidated practice within this tool called CMF (colours, materials, and finishes) [33], which is presented as a classification technique that organises the collected materials into three main categories, namely, the colours, materials, and finishes of a sector. The fundamental idea behind this classification technique is that an analysis of the three categories can be used to draw the "aesthetic profile" of a sector. The definition of aesthetic profiles is extremely important, especially in sectors where the dynamics and logic of product purchase and adoption are mainly guided by symbolic, perceptive, and inspirational dimensions, linked to the form and "languages" of the products. However, its usefulness is related to sectors dominated by functional and technological purchase logic. Furthermore, the dispersion of boundaries between sectors (for example, the textile sector that goes beyond the fashion supply chain to reach the lighting, furniture, and habitat products sectors) is combined with cross-fertilisation processes, which hybridise technologies and production practices between different sectors, highlighting the increasingly marked existence of a close relationship between different product areas in relation to formal solutions, colours, finishes, and materials that tend to move more quickly and frequently from one product to another. This tool is a web application that processes a product dataset from a knowledge graph. The dataset is defined by the exploratory search engine, considering the content of the the-saurus considered for product cataloguing and the Getty AAT thesaurus showing the relationship between the products of the dataset

according to user-defined parameters, and represents them in a 2D or 3D space as a function of the number of parameters considered. Figure 9 shows two screenshots of the AVD Product Map prototype.

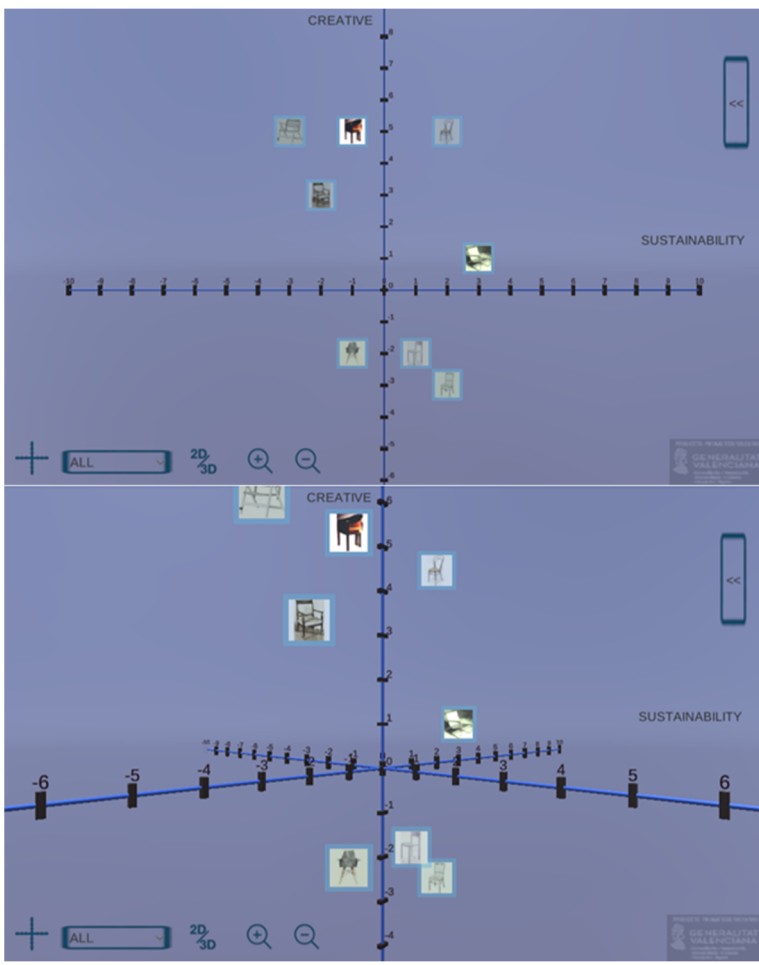

**Figure 9.** Screenshots of the AVD Product Map prototype, which displays the product dataset in 2D (**left**) and 3D (**right**), depending on the values of the user-defined variables.

### 4.3.2. Spatio-Temporal Map

AVD will develop a spatiotemporal map that allows the visualisation of data evolution in time and space and the detection of rare behavioural patterns by filtering and analysing the relationships between objects. This tool was the first result of the STEVO ontology and was created in the SILKNOW project [85]. A visualisation scene is an interactive map that allows dynamic data loading from a knowledge graph, and has many visualisation possibilities for data analysis. Figure 10 shows two screenshots of the spatiotemporal map used for the SILKNOW project. To use this tool in AVD, the content of the STEVO ontology must be adapted to project requirements. The AVD version needs to visualise products, companies, designers, documents, etc., and will use different time-reference units. Owing to the spatial distribution of the AVD dataset, new developments must be carried out to visualize large amounts of data located in the same place and to support dynamic load data pagination to avoid the deficiencies of the SILKNOW project version [91].

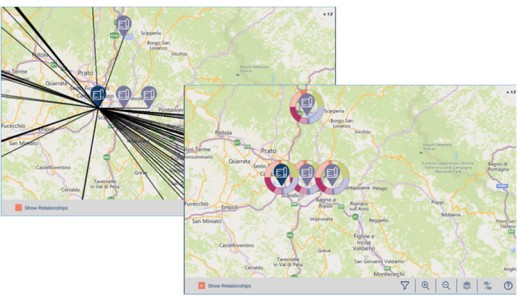
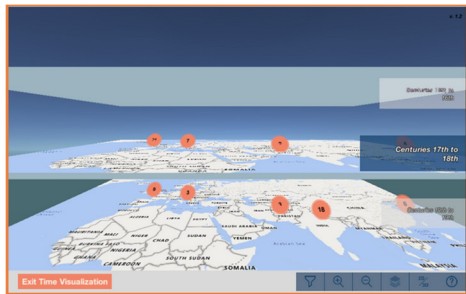

**Figure 10.** Screenshots of the spatiotemporal map used in the SILKNOW project (adapted from [87]. On the left represent the two ways of visualising the relationships between the objects, and on the right, the temporal variation of the spatial distribution of the objects.

### 4.3.3. Virtual Gallery

A virtual gallery is a useful tool for disseminating the content and experience of AVD. This will allow for an attractive and easy way to visualise and obtain information about it. In the market, there are many solutions for visualising museum content through virtual environments; however, there are no tools that allow dynamic data loading from a knowledge graph and take advantage of the data structure in interactive actions.

As a result, the virtual gallery designed and developed in this project will allow these features and will also suggest some visualisations of visitors as a function of their behaviour. For example, if a user remains looking at an object more than others and asking for information about it, the system could suggest that the user visit other virtual galleries with related objects with the same materials. In addition, the tool encourages the dissemination of AVD content, allowing the publication of videos and photos on social networks.

The content of the virtual gallery allows the user to reorganise information into different rooms organised by designers, companies, products, etc. The user will be able to change the hierarchy of the rooms in real time and will also be able to navigate naturally or jump to a specific room or object.

### 4.3.4. The Exploratory Search Engine

AVD is developing a knowledge graph based on the CIDOC-CRM model [92]. This knowledge graph contains digitised data and the semantic relationships between all objects. It was designed and developed to perform queries to obtain data from a knowledge graph for analysis. These data are exposed through dynamic forms or by using advanced semantic data visualisation tools. The end user could perform a simple text search and/or filter by specific filters (date, designer, location, etc.) The query result is displayed in a form in which part of the output can be hidden to facilitate content comprehension. The query results could also be visualised using semantic data visualisation tools, because the tools were integrated into the web search engine.

To achieve this objective, and given the complexity of the information to be handled, it is intended to design a web application, considering that it is the technological platform that best suits the needs of the project. Therefore, the tool will be a web application that must allow searching for information maintained in the knowledge graph, defined by an ontology based on CIDOC-CRM [93,94], which represents the necessary data model. The development of a computer support for the knowledge graph of a project is essential. In this study, the Openlink Virtuoso data virtualisation platform was installed on a server with the CentOS operating system. This platform is designed to maintain information in a knowledge graph that integrates the reasoning and inference functionalities. The future of the platform is guaranteed by a design that allows for high scalability and offers high-availability extensions. The Erlangen CRM/OWL implementation of the CIDOC-CRM conceptual model was extended and implemented on this platform. This platform

was selected by analysing the content of papers comparing the most used platforms for knowledge graph support and management [95–97].

Additionally, a knowledge graph population tool (KGCE) has been developed. This tool periodically processes the content of word-based websites, new data, and updates. This tool makes it possible to define a mapping between the source of the data and knowledge graph. If the data are linked to a thesaurus, as in this case, the link is established automatically. It also processes descriptions using artificial intelligence techniques to look for new relationships not present in the source data, and the documentation and cataloguing systems are part of an interconnected work with the development of technological tools. In addition, the web application must allow the selection of products from the graph and creation of a Product Map, which is a professional tool in the world of design. Along with the above, the content of the tool called "Design Memory" will also be shown through this platform.

### 4.4. Analysis

AVD applies comparative evaluation through benchmarking. This is a management tool for improvement through a comparative evaluation and in-depth structural analysis of successful practices carried out by companies, generally in the same field. Therefore, benchmarking consists of evaluating and analysing the processes, products, and services of other leading companies to conduct a comparative study and take them as a reference point for future strategies. This benchmarking analysis aims to provide a "snapshot" of the situation of parts or whole sectors in comparison with a series of variables-descriptors in different areas of design activities, as well as value as a factor of innovation.

The AVD has planned a series of workshops and focus group to identify the current needs and trends of designers, not only that but also the School of Art and Design (EASD) will be part of these activities so future designers can also be part. These workshops aimed to obtain comparative evaluations and deepen the knowledge of the sector for professional purposes and research. Through the participation in and attendance at these events, we intend to gain knowledge about the needs and handicaps faced by users who invest their efforts in the analysis and comparison of products. In addition, a characteristic feature of this proposal is the heterogeneous nature of members' profiles, where personalities from different disciplines meet.

Hence, it will be used within AVD as a point of action by companies, designers, and innovation managers, concurrent in the development of new products, or the adoption of new strategies. It appears often as a continuous and systematic process set in motion by companies to control the trajectory and pace of the development of their competitive environment. This tool makes it possible to collect products, analyse them, and classify them through the preparation of networks that provide effective cataloguing criteria and indicators. Hence, it serves to observe, visualise, and classify marketing reality to be able to analyse it to take pre-project ideas and strategic orientations. Comparative Evaluation analysis can take place both in a product and in communication support, the elements of corporate identity, as well as in the processes that support them. Benchmarking is identified by the choice of a panel of indicators (predefined parameters) that normally have characteristics of objectivity and measurability. The specific objectives of the analysis guide the definition of the inclusion/exclusion criteria for the set of products/processes to be compared guided by the analysis's specific objectives.

### 4.5. AVD Project Mock-Up

As mentioned, AVD digitises and catalogues archive records from Valencian designers' donations, incorporating them with their proper metadata. These data will be analysed and processed to homogenise their content and auto-matically retrieve semantic information, and a knowledge graph is created to manage the analysed information. Additionally, the outputs of queries are depicted through different visualisation tools, a dynamic map layer that accounts for their spatial and temporal dimensions, showing representations of the

Valencian industrial arena, and analysing the inter-product relation through interactive map products. These holdings will also be deployed using immersive digital technology. On the other hand, general users and professionals will be able to know the history behind these archives thanks to Design Archive Memory, which will be the basis for a comparative evaluation tool that will serve to compare products, brands, catalogues, etc., deepening the knowledge of the relationships between user needs and business strategies. The general workflow of the AVD is explained below:

- Digitisation and Catalogation of Collection.
- Interviews records, editing and making available
- Data processing.
- Integration. All the data were integrated into the ontology.
- Visualisation and interaction.
- Dissemination

To do so, the spatiotemporal map (STMap), the Product Map, and the Virtual Gallery will be integrated as a plug-in into the web search engine, allowing spatial and temporal navigation and relating the different products by showing their related properties, and also in a 3D room. The mockup will serve for prototyping and discussion of the graphical interface for both the control of these tools and their integration into the Web. The interface must show the navigation controls, as well as the visualisation selection controls of the time slot and its different partitions, as well as for the 3D room of the Virtual Gallery. Figure 11 shows the first version of the mock-up, where (a) corresponds to a screenshot of the main menu as seen in a user's browser, (b) screenshot to the search menu as seen in the user's browser, and (c) screenshot of the space-time map mock. Screen "map" of the mock tool; (d) Screenshot of the relational map mock.

In summary, the digital era has placed a wide range of tools at the service of doctoral and creative collections–and specifically in the field of conservation and innovation–to better enable conservation, management, and dissemination. Making design archives intelligible to society is now imperative, especially considering that culture is a universal right as part of the historical legacy. In addition, information technologies are fundamental in the transmission and preservation of collective memory; they also increase knowledge thanks to the generation of tools capable of serving as a motor for innovation and fostering the development of the economic and productive sectors associated with design.

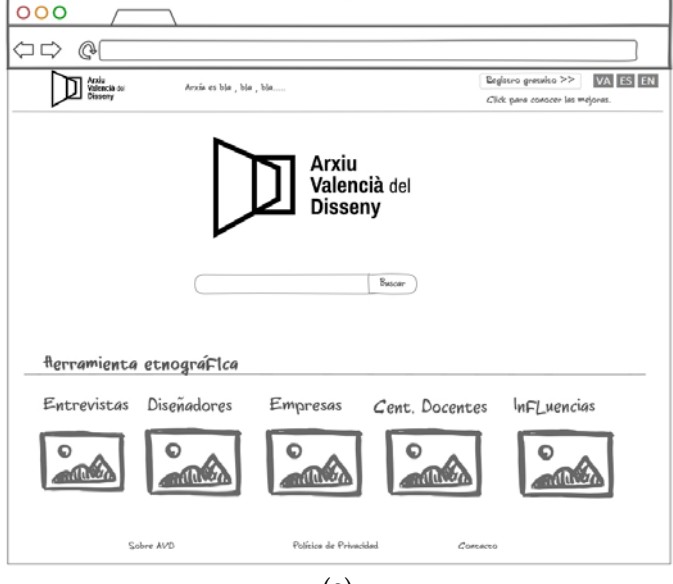

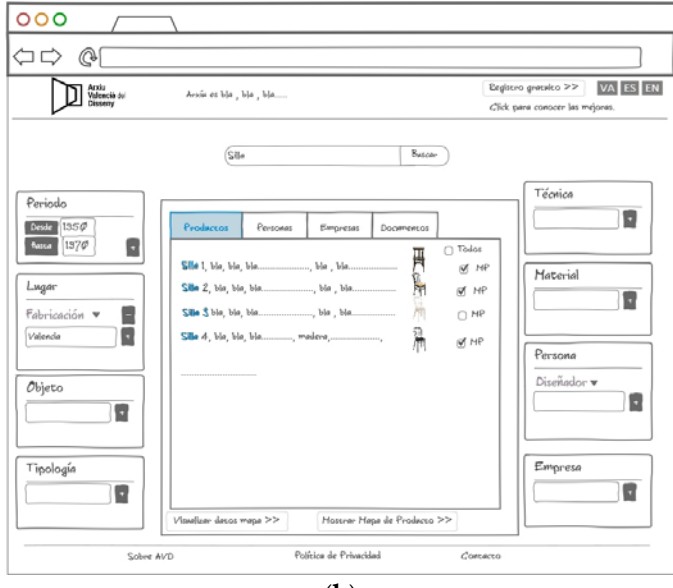

(a)　　　　　　　　　　　　　　　　　　　　　　　(b)

**Figure 11.** *Cont.*

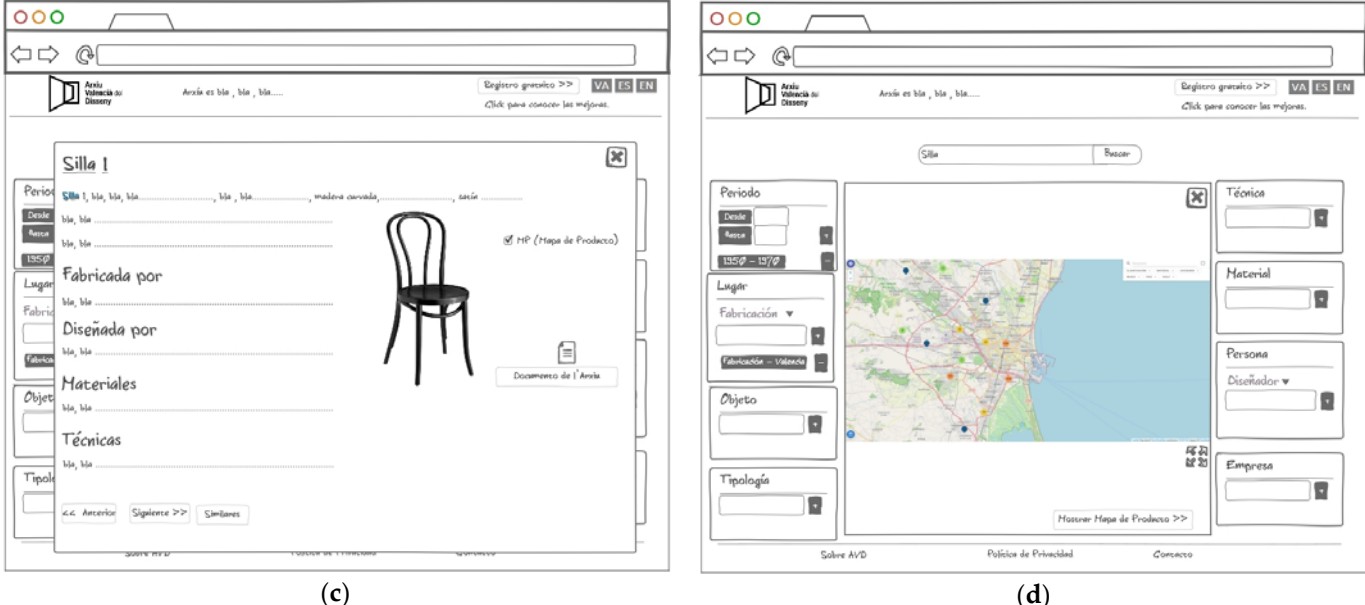

**Figure 11.** (**a**) Screenshot of the main menu as seen in a user's browser; (**b**) Screenshot of the search menu as seen in the user's browser; (**c**) Screenshot of the space-time map mock. Screen "map" of the mock tool; (**d**) Screenshot of the relational map mock.

### 4.6. Dissemination and Open Access

UNESCO set the preservation of "culture and cultural pluralism as tools for more resilient and peaceful societies" [50], as central social qualities in resistance against old and new crises. The AVD contributes to raising awareness of culture as the fourth pillar of sustainable development (in addition to the traditional three, environment, economy, and society) and to the EU Council's understanding of cultural heritage as a "strategic resource for a sustainable Europe" [51].

In this sense, AVD strengthens the uptake of research and innovation in society by enhancing co-creation strategies through archival dissemination. There is a five-fold helix to draw up proposals linked to the archive's collections. The bottom-up approach has proven particularly successful in reaching all social groups, including the most disadvantaged. This will be achieved by involving young designers in the project activities through activities created specifically for them, and working with the EASD has already proved to have some positive results, especially when working with heritage and innovation [50].

On the other hand, the AVD's collection and data collected through this project will be made widely and freely accessible to all interested parties to maximise project innovation potential and make this data useful for European citizens, institutions, and companies beyond the basic research and knowledge generation horizon. The digital data released by the AVD into the public domain will provide opportunities for new creative projects in different sectors, as fully described in the impact section below. By making this content available, we can create value and facilitate new business opportunities and concepts. Creative industries and GLAMs (galleries, libraries, archives, and museums) are experiencing a digital transition that involves shaking up traditional models, transforming value chains, and calling for new business models. AVD offers technology that can be especially useful for design companies who want to get inspiration for historical designs, allowing them to reuse this digitised heritage content as a design source and inspiration for new products and explore new business models for the company. Indeed, some of the project's results will have great indirect potential for industrial and commercial developments supporting new creative approaches that could turn into innovative services, leading to a change in traditional business models that build upon design. For instance, current designers will

be able to use the produced maps to evaluate the state of a concept, product, project, or benchmarking as a tool for adopting new strategies in their competitive sectors.

Similar points can be made about the project's potential interest in larger manufacturers, designers, 3D printing companies or the tourism industry. These sectors are already represented as collaborating institutions, thus guaranteeing an orientation towards possible indirect commercial and business applications of the results. In this sense, for the tourism industry, the tools and content created by the project, for example, will be instrumental in adding value to the new European Bauhaus. This encourages the development of related innovative tourism services, which could even be delivered via mobile devices, and supports the tourism promotion of GLAM institutions, considering the varied interests of cultural tourists interested in heritage design.

To do so, it is essential to incorporate different Valencian social agents into the process of enhancing the value of Valencian design. The co-creation of strategies for both the action and dissemination of the design archive collection is essential if it is to reach Valencian society, especially those groups that tend to remain outside cultural circuits. Above all, work is being done in a participatory manner on all issues related to the dissemination of the results of this research so that social agents are involved from the outset and collaborate to create a solid and clearly defined image of Valencian design. Moreover, it fosters industrial competitiveness, innovation, and technology but follows Europe's values, such as diversity or social equality.

On the other hand, related to our open access policies, the AVD is aware of the digital transformation that GLAMs are experiencing to make accessible and understandable for citizens [98,99] and how the practical implementations of the digital transformation vary greatly from one context to another, such as ownership and funding models of those collections, intellectual property rights, information resources (be they catalogues, inventories, mere accession lists, or any kind of intermediate models), and their availability in digital formats [100]. Digital transformation in all types of museums, especially due to the need of provide access to their collections, seems to have increased because of the COVID-19 crisis [101–105]. Indeed, open access to culture is a transformative concept that has profound implications for how we engage with and share our cultural heritage. Open access to culture breaks down barriers, ensuring that everyone, regardless of their background or location, can access and benefit from our shared cultural heritage. This promotes inclusivity and a more equitable distribution of knowledge to democratise access. In addition, by digitising and making cultural materials openly accessible, we can increase the chances of preserving them for future generations. Open access helps safeguard heritage from physical deterioration, loss, or destruction due to various factors [106–108]. Moreover, when artists, educators, and creators have access to cultural resources, they can draw inspiration from diverse perspectives, historical artefacts, and traditional knowledge. This has led to the creation of new works, educational materials, and innovative applications.

In this regard, the Berlin Declaration [109] stipulates that authors and holders of intellectual property must guarantee that all users are free, irrevocable, and worldwide access to scholarly work. This change speaks to the importance of knowledge reuse, but establishes an obligation to appropriate authorship.

AVD understands that we have different types of archives following different regulations. First, the Spanish Intellectual Property Law states that the term of the exploitation rights of the work is the life of the author and seventy years after his death, in which cases the author or its heirs will decide how much (if any) of his/her work wants to be Open Access. Second, we will follow Valencian Law which is practically identical to Spanish law. Nonetheless, the AVD research team has experience dealing with Intellectual Property Rights (IPR), as we have already dealt with similar issues in the SILKNOW project [91] which resulted in the development of a series of Best Practices for small and medium-sized museums who wish to access the digital arena [110]. n this sense, we understand that there is a wide range of intellectual property licences available. It is not just either the public domain or "all rights reserved" as Creative Commons licences provide many lay-

ered, nuanced variants, fit for most cultural institutions, and in our case, we will use them according to the desire of each donor. Furthermore, institutional guardianship, recognition of authorship, or any other moral rights will always be respected. Any rights associated with the object or its derivative data will be recognised and made visible to users through the different available open-access licences.

Finally, for end-users, AVD displays an interesting opportunity to understand the challenges of these types of heritage goods. In this sense, we are already dealing with it by means of focus groups and surveys, where we are asking designers, companies, and other relevant stakeholders how would they wish to access and/or give their work. The results will be an end-user policy for these types of archives.

## 5. Conclusions

For UNESCO, cultural heritage is becoming increasingly important to society. In this process, increasing modernisation and mutations in the dimension of social change and transformation are the keys to a new process, which includes greater socialisation of cultural heritage. The heritage of design culture is rich and falls within UNESCO's definition of integral heritage. In 2005, the FARO Convention on the Value of Cultural Heritage for Society highlighted the social and economic benefits of sustainable development [111].

On the other hand, UNESCO understands cultural and creative industries related to cultural heritage as those sectors of organised activity whose main purpose is the production or reproduction, promotion, dissemination, and/or marketing of goods, services, and activities with cultural, artistic, or heritage content. This approach emphasises the material good, the object as the final product of the creative or artistic process, and the channels of dissemination or the design process itself. Far from a homogeneous world, kaleidoscopes range from industrial processes to craftsmanship, traditional knowledge, design, and art. The main challenge of a set of realities is to strengthen local capacities, access international markets, protect cultural heritage, and preserve it [112,113].

In this process, cultural heritage linked to design is strongly linked to creative communities and associated with symbolic, material, and immaterial registers. It stands out for its uniqueness and the binding character of different heritage realities, and often remains linked to the present as a living heritage and to the processes of social transformation linking creativity and industry [114]. However, it can also become an important driver of local sustainable development through cultural tourism and as a source of inspiration for the creative industry, emerging at the European level as one of the few living heritages that currently maintain strong links with a cultural heritage that integrates a whole series of cultural and natural assets of different nature.

There are still issues concerning technical aspects of the sharing of digital archives; work needs to be done to understand the relationships between the digital and physical elements of archives, the connection between ownership and creative steps, physical and digital objects, and intergenerational play. It is important to note that there is no single size that fits all. Some archives and collections may struggle with an open access approach. Others may want to differentiate themselves to protect their assets. The small companies' archives were kept jealously for inspiration. They are used to train new generations of designers in their brand identity. New brands have developed archives that hope to capitalise on their future success, while others belong to companies that are no longer operating and are part of the post-industrial heritage. Some of these are collections that are kept in boxes, waiting to be catalogued. Some archives have important historical value and can shed light on historical periods and distinguished figures, such as fabric collections designed by Leonardo da Vinci. Some archives are in the public domain or in the process of being acquired through public collections (e.g., Museo del Tessuto, Prato, Italy). Fully digitalised archives can be accessed by users on creative commons or open-access licences, whereas others do not have the necessary legal structures to share content or provide access to users. Others are part of the cultural heritage of local communities but lack the mechanisms to return benefits to the community and are subject to free riding. In

addition to the lack of understanding of values and functions, there are issues related to the management and protection of physical objects in relation to distinct types of access and use (general public, designers, and purchasers), sharing of images or information for use, and reuse. Some of these objects risk losing their oblivion. Others are vulnerable to cultural and IP appropriation. In particular, within the intersection between a physical object and its digital representation, there are many issues concerning ownership, artistic creativity, technical innovation, and how these can be shared in a competitive and global market, such as in the fashion industry. Unlike artistic creations, fashion objects do not enjoy the same level of protection as cultural heritage objects do.

There is no updated survey on the modes of access and use from the perspective of archive holders and users. There was no map of the respective values or aspirations. The experience of the Creative Works project suggests that the boundaries among customers, users, and designers are blurred. There is no comprehensive map of the many types of fashion heritage archives, or their values and functions. As the NY Times asked, 'Should these Clothes be Saved?. . . But where, exactly, does their value lie?' (NYT 29/04/2012; Friedman, Vanessa). How can issues of their value facilitate the access and use of a global audience of users to support their protection and the community? What role does value play in the way stakeholders access and use fashion heritage sites?

The use and reuse of heritage brings a deeper awareness of local identities. However, Italy is built on fashion (Claudio Marenzi on the value of fashion, Google Arts and Culture). The use of heritage enhances its value, making it a 'living heritage', which in turn enriches, defines, and strengthens awareness of individual and collective identities [115]. Values associated with cultural heritage change according to society and time; however, the most accepted ones must be the intrinsic (the cultural object itself), instrumental (the benefits that society obtains from it), and institutional (education, preservation, and participation) values [116]. As noted, conservation and the values associated with cultural heritage lie in the interrelation between society and cultural heritage. This methodology, in which the community takes an outstanding place, comes from the promotion of talent, knowledge, and management of collective intelligence, especially in traditional arts and crafts, such as fashion, where traditional knowledge is essential to preserve it. Access is essential for raising cultural heritage awareness. In this sense, the reuse and use of heritage, thanks to participatory culture or digital access, allows the democratisation of cultural heritage by connecting people with cultural assets beyond their own reality. At the same time, it encourages the creation and co-creation of new narratives thanks to greater dissemination and interaction with the community, trans-forming cultural experiences, and giving greater value to digital data, generating social, cohesive, and economic benefits.

In 2011, the European Union charged a Report on bringing Europe's Cultural Her-Sitage online, highlighting the need for urgency to secure European heritage to younger generations and how cultural institutions can provide access without enduring the copyright of these institutions. This project can articulate a knowledge management system for the development of new products through the knowledge of free access to the heritage of the design culture. Since the growing conviction that knowledge constitutes the main asset of all types of organisations and actors (institutions, SMEs, ICC, creators, designers, etc.), and being complex knowledge management in practice, AVD assumes the challenge of generating structures and methods so that the paradox can be overcome and generate innovation-inducing results.

Most of the policies of the most developed countries in the integration of design in their productive and service sectors (Korea, Denmark, Ireland, Finland, Sweden, Norway, New Zealand, England) place emphasis on design as a strategic tool for economic progress, the improvement of competitiveness, the development of companies, and the creation of new business opportunities [117]. Without forgetting the issues of national identity, the image of the country or region, and the possibility of branding the country brand. Likewise, the strengthening of the design research sector and the exchange of knowledge through the establishment of international networks are among the objectives of most design policies

in developed countries. The AVD will fulfill a key role in achieving these objectives, by providing an analog and digital space where in-formation can be exchanged, good design models identified, promoted through exhibi-tions, meeting their authors, establishing contacts with similar centers in other countries.

It is specifically oriented to the product development process because of the complexity involved in the management of knowledge in the same, aiming to be an effective and efficient resource, achieving it by favouring its use and making it available to all the actors involved. Thus, it has become an instrument for the positioning of design as a strategic factor in the local innovation system through the implementation of specific guidelines for its practices and the achievement of tangible results by all the actors of the innovation system, such as an instrument of support for the cultural and scientific policies of the design system [118]. Equally, it can become a basic instrument for the development policies of the design system, interacting with different disciplines (marketing, productive technologies, digital technologies, design, art, etc.) from its centrality within the innovation system. One of the most identifying assets of AVD is its content specificity: design, all its areas, practices, and manifestations. Practice devoid of rigid systematizations, on the contrary, equipped with interpretative models capable of "explaining" the dynamics and the relationships that exist between the different variables of the problems. Finally, this philosophy aims to generate an interpretive model for the project culture. Project culture aims to create better situations than the existing ones, so it is difficult to codify a practice, model, and unique and shared conceptual scheme. The codification of a model (simplification of reality) naturally involves promoting replicability and reproducibility, synonymous with standardisation and uniformity [119]. These principles collide head-on with the nature of design, because within the culture of the project, there are no automatic deductive mechanisms, and there are no models that provide closed solutions based on strict rules. On the contrary, AVD as an agent of innovation, through its digital tools, intends to make open access a material that provides extensive knowledge and various interpretive models capable of generating the relationships that exist between the different variables of the problems and from that interpretive approach to become a process facilitator and activate the development of the project culture. In this sense, the AVD project is aligned with the principles of the new European Bauhaus, presented by the President of the European Commission, Úrsula Von der Leyen, in the State of the Union speech of September 2020 as a determined action to improve the lives of citizens. With this initiative, an appeal is made to all European citizens to reflect on the spaces in which our lives develop. This European project shows that culture is a common link among different European countries [120].

The main open access and specific platforms used in this project, OpenLink Virtuoso and Three. JS, are distributed under the GNU General Public Licence (GPL) version 2 and MIT licence, respectively. Licencing requirements are appropriate for an AVD project. Both platforms were developed more than ten years ago and have a large number of users and documentation repositories. They might not be maintained in the future, but it has been difficult for this to happen in recent years. However, the design and development of STEVO allows the replacement of the graphical platform. The same applies to OpenLink Virtuoso software (v.7.2.10), which can easily be replaced in the project.

This differentiating nature is related to what is often identified as the final objective of a project's activities: innovation. Among its potential, AVD highlights the ability to interpret the legacy it protects through the digital tools it formulates.

**Author Contributions:** Conceptualization, M.G. and E.A.; methodology, M.G.; software, J.S.; valida-tion, J.S., A.L., Á.S., V.P., E.A. and M.G.; formal analysis, A.L., Á.S., V.P., E.A. and M.G.; investigation, A.L., Á.S., V.P., E.A. and M.G.; resources, V.P.; data curation, J.S.; writing—original draft preparation, J.S., A.L., Á.S., V.P., E.A. and M.G.; writing—review and editing, A.L. and M.G.; visualization, A.L. and M.G. supervision, E.A. and M.G.; project administration, E.A.; funding acquisition, E.A. All authors have read and agreed to the published version of the manuscript.

**Funding:** This research was funded by the Generalitat Valenciana, Conselleria d'Innovació, Universitats, Ciència i Societat Digital, grant number Prometeo/2021/001. Solbes is supported by the Spanish government postdoctoral grant María Zambrano. Gaitán is funded is supported by the Spanish government postdoctoral grant Margarita Salas under grant No MS21-138.

**Data Availability Statement:** The current data presented in this study are openly available in https://www.uv.es/avd (accessed on 25 April 2023).

**Conflicts of Interest:** The authors declare no conflict of interest.

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
