# Peer review of "Technological Tools for the Conservation and Dissemination of Valencian Design Archives"

_heritage, doi:10.3390/heritage6090319_

Round 1
Reviewer 1 Report
The contribution describes a very interesting project and is well presented, but in my opinion it needs to devote more space to Open Access. In fact, it is not enough to state that you want to use Open Access, especially since it is a design archive and as has been explained, the origins of the data may be different.
To be indicated:
what are the possible limitations and difficulties of Open Access?
What are the possible licences to be applied?
Will user policies be drawn up?
Furthermore, in the analysis of the platforms taken as a model, a summary table with the characteristics presented would be useful.
An overall mockup of the project is missing
Quality of English Language is good
Author Response
The contribution describes a very interesting project and is well presented, but in my opinion it needs to devote more space to Open Access.
We want to thank the reviewer for his/ her kinds comments.
- In fact, it is not enough to state that you want to use Open Access, especially since it is a design archive and as has been explained, the origins of the data may be different.
To be indicated:
what are the possible limitations and difficulties of Open Access?
What are the possible licences to be applied?
Will user policies be drawn up?
We have addressed these issues in section 4.6, which is now called “Dissemination and Open Access”. Also we have added the possible limitations of the Open Access tools used in the AVD project in the conclusions sections where the licence of the libraries is also described.
- Furthermore, in the analysis of the platforms taken as a model, a summary table with the characteristics presented would be useful.
Many papers were consulted in order to evaluate the knowledge graph tool appropriate to the project goals. Some of these papers were not included in the last version. In this version we include some lines describing the process and the references to these papers.
A table of different web graph representation libraries is included in the paper, as well as the library chosen and why.
- An overall mockup of the project is missing
We have added a new section with some images and explanations about how the project will be seen by end-users in section 4. “AVD Mock-up"

Reviewer 2 Report
The presented work is interesting and I also appreciated that the paper was clear, easy and pleasant to read. However, there are some weaknesses in the content that should be addressed to improve the significance of the reported work. Here below some suggestions that may be of help.
1) The Introduction should contain a more effective description of the research questions / innovations that the AVD project addresses and how it contributes to advance the state of the art.
2)Section 2 (State of the art) should be expanded on subsection 2.1.
3) Section 3 needs more specific title regarding the concept of the section. Also I can see subsection 3.2. Where is 3.1?
4) In Section 4, please consider adding another thesaurus related publication: Konstantakis, Markos, et al. "ACUX Typology: A Harmonisation of Cultural-Visitor Typologies for Multi-Profile Classification." Digital 2.3 (2022): 365-378.
Acierno, M.; Cursi, S.; Simeone, D.; Fiorani, D. Architectural heritage knowledge modelling: An ontology-based framework for conservation process. J. Cult. Herit. 2017, 24, 124–133.
Author Response
Dear Reviewer 2,
On behalf of the authors, we wish to thank you for your generous comments on the manuscript. We have edited it to address your concerns.
1) The Introduction should contain a more effective description of the research questions / innovations that the AVD project addresses and how it contributes to advance the state of the art.
We have added a short paragraph in the introduction where we specify our research questions and how we contribute to the advance of the state of the art.
2)Section 2 (State of the art) should be expanded on subsection 2.1.
We have expanded subsection 2.1 with more references and more examples of archives using digital technologies. We especially explained how we are going beyond the state of the art.
3) Section 3 needs more specific title regarding the concept of the section. Also I can see subsection 3.2. Where is 3.1?
We have changed the section title and rearranged the section as there was a typo on section 3.2, it was section 3.1
4) In Section 4, please consider adding another thesaurus related publication: Konstantakis, Markos, et al. "ACUX Typology: A Harmonisation of Cultural-Visitor Typologies for Multi-Profile Classification." Digital 2.3 (2022): 365-378.
Acierno, M.; Cursi, S.; Simeone, D.; Fiorani, D. Architectural heritage knowledge modelling: An ontology-based framework for conservation process. J. Cult. Herit. 2017, 24, 124–133.
We have added some references related to thesaurus, including those referred by the reviewer.
We believe that the manuscript is now suitable for publication in Heritage.
